# First-Principles Calculation of the Bonding Strength of the Al_2_O_3_-Fe Interface Enhanced by Amorphous Na_2_SiO_3_

**DOI:** 10.3390/ma15134415

**Published:** 2022-06-22

**Authors:** Shaosheng Wei, Xiaohua Yu, Dehong Lu

**Affiliations:** 1Faculty of Materials Science and Engineering, Kunming University of Science and Technology, Kunming 650093, China; sswei73112022@126.com (S.W.); xiaohua_y@163.com (X.Y.); 2Yunnan Vocational College of Mechanical and Electrical Technology, Kunming 650203, China

**Keywords:** particle-reinforced composites, Fe-amorphous Na_2_SiO_3_-Al_2_O_3_, bonding action, first-principles calculations

## Abstract

In this paper, the interfacial adhesion work (W*_ad_*), tensile strength, and electronic states of the Fe-amorphous Na_2_SiO_3_-Al_2_O_3_ and Fe-Al_2_O_3_ interfaces are well-investigated, utilizing the first-principles calculations. The results indicate that the Fe-amorphous Na_2_SiO_3_-Al_2_O_3_ interface is more stable and wettable than the interface of Fe-Al_2_O_3_. Specifically, the interfacial adhesion work of the Fe-amorphous Na_2_SiO_3_ interface is 434.89 J/m^2^, which is about forty times that of the Fe-Al_2_O_3_ interface, implying that the addition of amorphous Na_2_SiO_3_ promotes the dispersion of Al_2_O_3_ particle-reinforced. As anticipated, the tensile stress of the Fe-amorphous Na_2_SiO_3_-Al_2_O_3_ interface is about 46.58 GPa over the entire critical strain range, which is significantly greater than the Fe-Al_2_O_3_ interface control group. It could be inferred that the wear resistance of Al_2_O_3_ particle-reinforced is improved by adding amorphous Na_2_SiO_3_. To explain the electronic origin of this excellent performance, the charge density and density of states are investigated and the results indicate that the O atom in amorphous Na_2_SiO_3_ has a bonding action with Fe and Al; the amorphous Na_2_SiO_3_ acts as a sustained release. This study provides new ideas for particle-reinforced composites.

## 1. Introduction

As is generally known, ceramic-particle-reinforced composite materials significantly improve the hardness and wear resistance of metallic materials [1,2,3,4]. Among particle-reinforced materials, Al_2_O_3_ particle-reinforced has received a lot of attention due to its outstanding physicochemical properties, such as high melting point, hardness, wear resistance, and so on [5,6]. However, Al_2_O_3_ is brittle and its thermophysical expansion coefficient is markedly different from that of metallic materials, which seriously limits its popularization and application [7]. In addition, the weak bonding strength leads to the Al_2_O_3_ particles peeling away from substrates during the friction and wear processes. Therefore, the development of innovative ways to improve the bonding strength between the Al_2_O_3_ particulate reinforcement and the Fe substrate is the key to addressing the issues.

Various strategies have been proposed to improve the bonding strength between the Al_2_O_3_ particulate reinforcement and the Fe matrix. The first strategy is that the addition of metal elements such as Ti and Ni promotes the penetration of Fe atoms into the Al_2_O_3_ particles [8,9,10]. The second one is that silicates are filled between the Al_2_O_3_ coating and the substrates in a sandwich-like structure [11]. In fact, the interfacial bonding strength mechanism of the solid/solid interface at the atomic and electronic levels is difficult to investigate owing to the complexity of the interface region in terms of tests. Advanced detection devices such as high-resolution transmission electron microscopy and focused ion beams only monitor and analyze the local atomic distribution, which is the moment state of atoms. The conventional thermodynamic and kinetic analyses are not capable of simulating the atom-atom interactions which are essential to understand the interfacial bonding strength mechanism. Therefore, the first-principles calculation is conducted to investigate the amorphous modification layer, which is the key to dealing with the issues [12].

The first-principles calculations based on density functional theory have been used successfully for the solid-solid interfaces, which indicate the atoms and electric and mechanical structures. Li et al. [13] found that compared with the Al-terminated structure of the Al_2_O_3_-Fe interface, the O-terminated structure shows better tensile properties, in which the critical strain of the O-terminated structure is 9.5% higher than that of the Al-terminated structure, and the ideal tensile strength is 3.5-fold that of the Al-terminated structure. In a further study, Tran [11] found that the Fe atoms on the surface have three covalent bonds with the below oxygen atoms to form covalent bonds with other oxygen atoms. Zhao et al. [14] reported that the wear resistance of the Fe matrix can be improved by doping Ti and Ni. The improvement in wear resistance is a result of the elements being added, which strengthens the interfacial bonding strength between the carbide and the substrate. Wan et al. [15] showed that the silicate can react with the Fe_3_O_4_ surface to form a eutectic mixture, Fe_2_SiO_4_, which improves the friction reduction and anti-wear performance. To the best of our knowledge, research on the addition of amorphous Na_2_SiO_3_ to the Fe-Al_2_O_3_ interface to improve the bonding strength of Fe-Al_2_O_3_ has not been reported. In the research on alumina-ceramic-reinforced iron matrix composite interface, amorphous sodium silicate was found, which makes the interface bond more tight [16]; we speculate that amorphous sodium silicate will have an effect on the interface bond.

This theoretical study of adding water glass to solve the interfacial bonding problem of ceramic-particle-reinforced steel (iron-based) composites has some theoretical guidance. In this work, the adhesion work, tensile failure, and electronic structure of the Fe-Al_2_O_3_ and Fe-amorphous Na_2_SiO_3_-Al_2_O_3_ interfaces are examined using first-principles calculations. Firstly, the Fe-amorphous Na_2_SiO_3_-Al_2_O_3_ and Fe-Al_2_O_3_ interfaces are meticulously constructed. Secondly, the first-principles stretch calculation method is utilized to explore the interaction of the interfaces under stress and demonstrate the wear resistance during service processes. Finally, the origin of excellent properties is investigated from the electronic angle point to confirm the excellent properties of the binding mechanism for the Fe-amorphous Na_2_SiO_3_-Al_2_O_3_ interface. Our work provides a totally new strategy that enhances the bonding strength of the Fe-Al_2_O_3_ interface by doping amorphous Na_2_SiO_3_ to obtain high-performance wear-resistant materials.

## 2. Calculation Details

### 2.1. Calculation Parameters

The first-principles calculations are implemented using the Cambridge Sequential Total Energy Package (CASTEP) code [17,18]. The simulations are based on density functional theory (DFT) under plane-wave basis, periodic boundary conditions, and the projected augmented wave approach (PAW) [19]. The Generalized Gradient Approximation (GGA) in the Perdew–Burke–Ernzerhof (PBE) is employed to describe the exchange-correlation energy [20,21]. The 8 × 8 × 8 and 8 × 8 × 1 Monkhorst–Pack [22] of *κ*-point are used to sample the Brillouin zone of the bulks and interfaces, respectively, and the plane-wave cutoff energy is 380 eV. Ultrasoft pseudopotentials are applied to all the elements [23]. Moreover, the force and energy convergence precision are 10^−6^ eV and 0.01 eV Å^−1^, respectively.

### 2.2. Selection of Models

The crystal structure of γ-Fe, Na_2_SiO_3_, and α-Al_2_O_3_ is shown in Figure 1. The γ-Fe (a = b = c = 3.439 Å) belongs to FM-3M (No. 225) space groups and the Na_2_SiO_3_ (a = 6.027 Å, b = 10.487 Å, c = 4.784 Å) and α-Al_2_O_3_ (a = b = 4.814 Å, c = 13.156 Å) belong to Cmc2-1(No. 36) and R-3 C (No. 167) space groups, respectively, as shown in Table 1. In this work, the Na_2_SiO_3_ is amorphous; the Na_2_SiO_3_ occurs as a structural transformation from crystalline to amorphous phase by forcite mode in the materials studio. The crystal structure transformation is more beneficial to simulate an approximation to the real crystal state. To meet the rigor of the research, the interface structure consistent with previous reports is selected in this work [13]. For the crystal surface and the layer, the γ-Fe (111) and α-Al_2_O_3_ (0001) surfaces with the lowest energy are selected in exponential crystal planes [24,25]. When the Fe (111) and Al_2_O_3_ (0001) surfaces are 7- and 5-layer, the surface convergence and the stability test are satisfied. The γ-Fe (111) combines directly with the α-Al_2_O_3_ (0001) by the O-terminated structure, and the amorphous Na_2_SiO_3_ is inserted into the middle between them, as shown in Figure 2. In addition, there are three stacking models for the O-terminated Fe-amorphous Na_2_SiO_3_-Al_2_O_3_ and Fe-Al_2_O_3_ interfaces, as shown in Figure 3. The ideal adhesion work (*W_ad_*) is calculated to judge bonding strength and scales as in [26,27]:(1)$$W_{ad}=\frac{E_{1}+E_{2}-E_{1/2}}{A}$$

As the model created has three interfaces, the details are as follows: the ideal adhesion work (Wad) is between the Fe-Al_2_O_3_ interface, where $E_{1}$ is the total energy of γ-Fe (111) and $E_{2}$ is represented as the total energy of α-Al_2_O_3_ (0001), respectively. $E_{1/2}$ is the total energy of the Fe-Al_2_O_3_ interface systems; A is the interfacial area. The interface is between Fe and amorphous Na_2_SiO_3_-Al_2_O_3_ systems, where $E_{1}$ is the total energy of γ-Fe (111) and $E_{2}$ is represented as the total energy of α-Al_2_O_3_ (0001) and amorphous Na_2_SiO_3_, respectively. $E_{1/2}$ is the total energy of the Fe-amorphous Na_2_SiO_3_-Al_2_O_3_ interface systems; A is the interfacial area. The interface is between Fe-amorphous Na_2_SiO_3_ and Al_2_O_3_ systems, where $E_{1}$ is the total energy of γ-Fe (111) and amorphous Na_2_SiO_3_ and $E_{2}$ is represented as the total energy of α-Al_2_O_3_ (0001), respectively. $E_{1/2}$ is the total energy of the Fe-amorphous Na_2_SiO_3_-Al_2_O_3_ interface systems; A is the interfacial area.

### 2.3. Method of Tensile Test

In the tensile test, a uniaxial tensile strain is applied to the chosen direction of the γ-Fe-α-Al_2_O_3_ and γ-Fe-amorphous Na_2_SiO_3_, as shown in Figure 4. The tensile test steps are given in the following general procedures: first, a small fixed strain is loaded in the chosen direction (in this work, the [001] direction); second, the crystals are elongated in the loading axis; third, the crystals are relaxed adequately. The above processes are repeated until the crystals are separated. The tensile stress (*σ*) and the strain (*ε*) are calculated from the formula [28]:(2)$$\varepsilon =\frac{L-L_{0}}{L_{0}}\times 100\;\%$$
(3)$$\sigma_{\left(\varepsilon \right)}=\frac{1+\varepsilon }{V_{\left(\varepsilon \right)}}\cdot \frac{\partial E_{\left(\varepsilon \right)}}{\partial \varepsilon }$$
where *L_0_* and *L* are the original and stretched length in the loading axis, respectively. $V_{\left(\varepsilon \right)}$ and $E_{\left(\varepsilon \right)}$ are the volume and total energy of the crystals related to the strain (*ε*), respectively.

## 3. Results and Discussion

### 3.1. Adhesion Work and Wettability

The wettability of Fe-amorphous Na_2_SiO_3_-Al_2_O_3_ and Fe-Al_2_O_3_ interfaces could be characterized utilizing the adhesion work. Figure 5 illustrates the adhesion work for the Fe-Al_2_O_3_ and Fe-amorphous Na_2_SiO_3_-Al_2_O_3_ interfaces as a function of the number of atomic layers. The interlayer adhesion work of most atomic layers does not reveal noticeable changes. However, the adhesion work of the Fe-Al_2_O_3_ interface increased suddenly to 10.46 J/m^2^ of the 7th layer (Figure 5a). It can be expected that this is due to the difference in the combination between Fe and Al_2_O_3_, and further mechanistic analysis is discussed in Section 3.3. Notably, the interlayer adhesion work at the positions between 8th and 12th for the Fe-amorphous Na_2_SiO_3_ interface is about 434.89 J/m^2^, which is about 10-fold that of the Fe-Al_2_O_3_ interface. In comparison to the Fe-Al_2_O_3_ interface, the stronger interlayer adhesion work contributed to the Fe-amorphous Na_2_SiO_3_ interface’s more stable bonding strength. The adhesion work at the composite interface was found to become higher after the addition of amorphous Na_2_SiO_3_ from theoretical studies. Practical experiments revealed that amorphous Na_2_SiO_3_ promotes bonding at the interface of ceramic-reinforced steel matrix composites [16]. Therefore, it is inferred that the addition of amorphous Na_2_SiO_3_ promotes the dispersion of Al_2_O_3_ particles and obtains a more uniform structural organization.

### 3.2. Tensile Test

The safety index and the mechanical properties of materials can be evaluated by the tensile test. Figure 6 presents the tensile stress related to strain of the Fe-Al_2_O_3_ and Fe-amorphous Na_2_SiO_3_-Al_2_O_3_ interfaces. The tensile strength of the Fe-Al_2_O_3_ interface increases gradually with the strain and reaches its maximum value of 31.36 GPa at the critical strain of 10%. Then, the tensile strength decreases sharply to zero as the strain is more than 10%, illustrating that the bonding strength for the Fe-Al_2_O_3_ interface is far from satisfactory. It is worth noting that Li et al. [13] reported that the maximum value of tensile strength is 31.16 GPa at the critical strain of 10% for the Fe-Al_2_O_3_ interface, which coincides with the calculation results and demonstrates the veracity of the calculation. The tensile strength of the Fe-amorphous Na_2_SiO_3_ interface increases slowly when the strain is less than 10%. This is because the amorphous structure exhibits excellent stress-release properties. When the strain is between 10% and 20%, the tensile strength is maintained at approximately 46.58 GPa. The reason for this is that the amorphous structure efficiently alters the atomic location and slows down the deformation when it is impacted by external forces. Obviously, it is another excellent property of the amorphous Na_2_SiO_3_ interface. Finally, the maximum tensile strength is 46.58 GPa for the Fe-amorphous Na_2_SiO_3_-Al_2_O_3_ interface at the strain of 20%, which is 1.5-fold that of the Fe-Al_2_O_3_ interface. In summary, the bonding strength of the composite materials is effectively improved as a result of the addition of amorphous Na_2_SiO_3_, which lays a good foundation for wear resistance during service.

### 3.3. The Electronic Origin of Adhesion Work and Wettability

Generally, macroscopic physical phenomena are associated with microscopic atomic states. To better understand the mechanical bonding of the atomic structure of the Fe-amorphous Na_2_SiO_3_-Al_2_O_3_ and the Fe-Al_2_O_3_ interfaces, the electronic origin of the two kinds of interfaces is investigated in detail. Figure 7 shows the geometrical structure of the Fe-Al_2_O_3_ and the Fe-amorphous Na_2_SiO_3_-Al_2_O_3_ interfaces. The O-terminated structure for the Fe-Al_2_O_3_ interfaces has higher adhesion work than the Al-terminated interfaces; however, the charge accumulation between the Fe- and Al-terminated structures is weak [13]. As a result, this work focuses primarily on the O-terminated structure for interfaces. Figure 8 presents the charge density and charge density difference of the Fe-Al_2_O_3_ interface at (111) crystal plane. It is obvious that the charge accumulation exists between O and Fe atoms, which reveals the chemical bonding formed between them. This conclusion is consistent with the previous study showing that the O-terminated interface is more stable than the Al-terminated interface in the Fe-Al_2_O_3_ interfaces. The charge density of the atoms in each crystal plane is studied to verify the accuracy of the results. Therefore, the charge density of the Fe-Al_2_O_3_ interface at (001) crystal plane is also analyzed. The charge density and charge density difference of the Fe-Al_2_O_3_ interface at (001) crystal plane are shown in Figure 9. It is clear that the results are comparable to those shown in Figure 8.

Next, the charge density and charge density difference of the Fe-amorphous Na_2_SiO_3_-Al_2_O_3_ interface are also taken up in discussion. Charge accumulation is obviously visible at the Fe-amorphous Na_2_SiO_3_ interface, as shown in Figure 10a. The charge density of the Fe-amorphous Na_2_SiO_3_ interface is greater than that of the Fe-Al_2_O_3_ interface, implying that the bonding strength is likewise greater than that of the Fe-Al_2_O_3_ interface. Furthermore, the charge density between the amorphous Na_2_SiO_3_-Al_2_O_3_ is also of concern. Figure 10b shows the charge density difference of the Fe-amorphous Na_2_SiO_3_-Al_2_O_3_ interface. There is a vast region of charge sharing zones among the Fe, Si, and O atoms, and the charge accumulation zone is large. Therefore, the chemical bonding existing between these atoms is nonpolar covalent [29]. However, for the amorphous Na_2_SiO_3_-Al_2_O_3_ interface, the charge accumulation is skewed to the O atom, which indicates that the polar covalence formed between the Si and O atoms. Concurrently, the charge density difference of the Fe-amorphous Na_2_SiO_3_-Al_2_O_3_ interface at the (001) crystal plane is similar to that at the (111) plane, as shown in Figure 11. The above results reveal that the chemical bonding of the Fe-amorphous Na_2_SiO_3_-Al_2_O_3_ interfaces is stronger than that of Fe-Al_2_O_3_ and the bonding strength is a significant improvement.

Furthermore, the hybridization of the electrons and chemical bonding at the interfaces are explored by utilizing the partial density of states (PDOS). Figure 12 presents the partial density of states (PDOS) at the Fe-amorphous Na_2_SiO_3_-Al_2_O_3_ and Fe-Al_2_O_3_ interfaces. The black solid line depicts s-states, the red solid line represents p-states, the blue solid line shows d-states, and the black dotted line defines the Fermi level. The PDOS of the d electrons of Fe and the p electrons of O bulges and exceeds zero at the Fermi level, as shown in Figure 12a. Therefore, the metallic bonding between the Fe and O atoms appears at the interfaces. Moreover, orbital hybridization exists between the Fe-d and O-p in the region of −6.2 eV to 0.5 eV, which reveals that a covalent connection is created. Therefore, a combination of metallic bonds and covalent bonds occurs at the interface. In the Fermi level region, the PDOS of the Al atom does not appear when compared to the Fe atom, implying that the Al chemical bonding is barely noticeable.

However, the PDOS of the Fe-amorphous Na_2_SiO_3_ interface shows strong chemical bonding between Si and O atoms, with the dominant contributions being the s and p electrons of Si and the p electrons of O. The O-p, Fe-d, and the Si-p, d state bulge up in the region between −10 eV and 0.5 eV. This suggests the formation of a strong covalent bond at the Fe-amorphous Na_2_SiO_3_ interface. The degree of p and d hybridization at the Fe-amorphous Na_2_SiO_3_ interface is higher than that of Fe-Al_2_O_3_. The hybridization between p and d orbitals determines the adhesion strength and stability of interfaces [28]. Therefore, the adhesion strength and stability of Fe-amorphous Na_2_SiO_3_ interfaces are stronger than those of Fe-Al. The s and p electrons of Al in the region between −10 eV and 0 eV also protrude, which demonstrates that the covalent bond formed at the amorphous Na_2_SiO_3_-Al_2_O_3_ interface is stable. Therefore, the adhesion strength of the amorphous Na_2_SiO_3_-Al_2_O_3_ interface is also strengthened. Collectively, the electronic origin of bonding strength is summarized by amorphous Na_2_SiO_3_ forming stable covalent chemical bonds with Fe and Al_2_O_3_ to act as a sustained release.

## 4. Conclusions

As a wear-resistant material, Al_2_O_3_ has received extensive attention. However, weak bonding strength causes Al_2_O_3_ particles to peel off from the substrate during friction and wear. In this study, adhesion work and tensile tests for the Fe-Al_2_O_3_ and Fe-amorphous Na_2_SiO_3_-Al_2_O_3_ interfaces are explored by applying first-principles calculations. The interlayer adhesion work for the Fe-amorphous Na_2_SiO_3_ interface is about 10-fold that for the Fe-Al_2_O_3_ interface. Meanwhile, the maximum tensile strength of the Fe-amorphous Na_2_SiO_3_ interface is 1.5-fold that of the Fe-Al_2_O_3_ interface, and the critical strain is 10% higher than that of the Fe-Al_2_O_3_ interface. Further, the charge density of the Fe-amorphous Na_2_SiO_3_ interface is greater than that of the Fe-Al_2_O_3_ interface, implying that the bonding strength is likewise greater than that of the Fe-Al_2_O_3_ interface. Therefore, this work provides a theoretical explanation for the effect of the amorphous Na_2_SiO_3_ on the bonding strength of the Al_2_O_3_-Fe interface.

## Figures and Tables

**Figure 1 materials-15-04415-f001:**
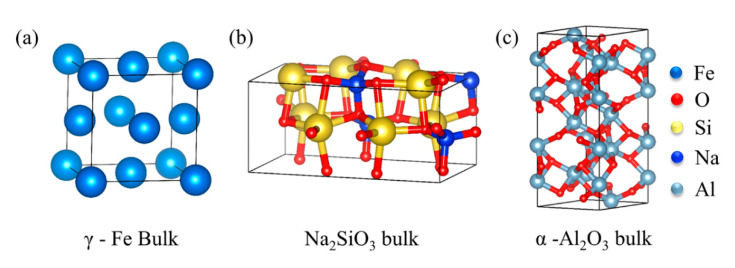
The crystal structure of (**a**) γ-Fe, (**b**) Na_2_SiO_3_, and (**c**) α-Al_2_O_3_.

**Figure 2 materials-15-04415-f002:**
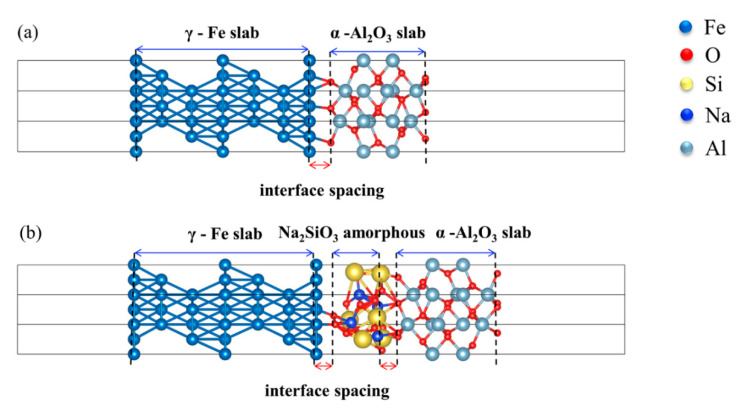
The model of the (**a**) Fe-Al_2_O_3_ and (**b**) Fe-amorphous Na_2_SiO_3_-Al_2_O_3_ interfaces.

**Figure 3 materials-15-04415-f003:**
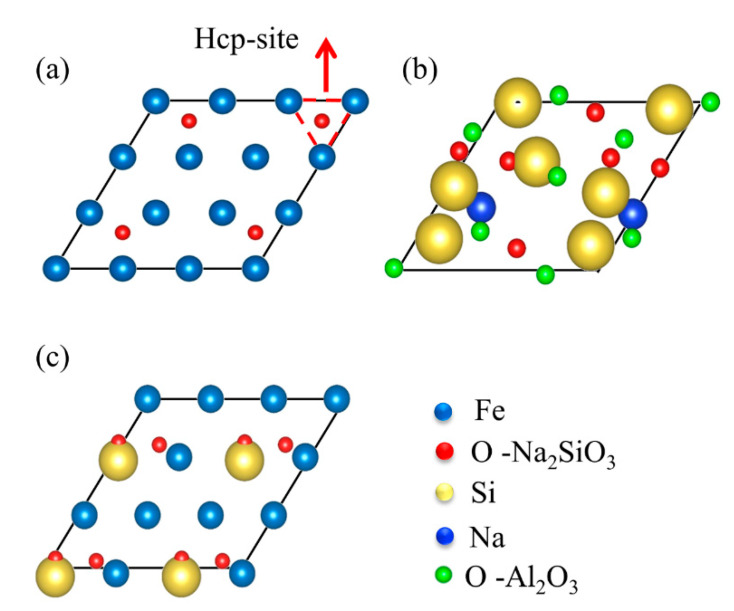
Three kinds of stacking sequences for the O-terminated Fe-amorphous Na_2_SiO_3_-Al_2_O_3_ and Fe-Al_2_O_3_ interfaces: (**a**) hcp, (**b**) hole, and (**c**) top.

**Figure 4 materials-15-04415-f004:**
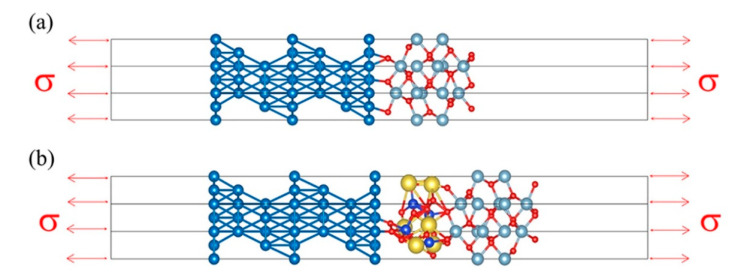
The tensile simulation model for the (**a**) Fe-Al_2_O_3_ and (**b**) Fe-amorphous Na_2_SiO_3_-Al_2_O_3_ interfaces.

**Figure 5 materials-15-04415-f005:**
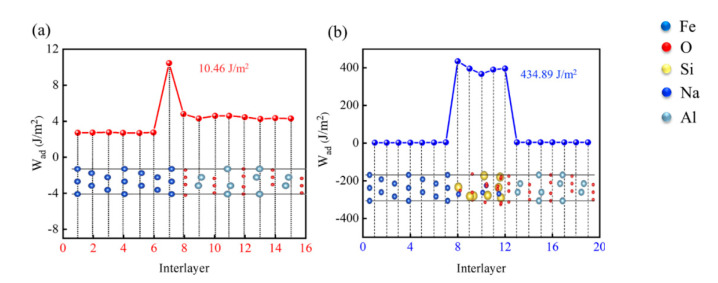
The adhesion work for the (**a**) Fe−Al_2_O_3_ and (**b**) Fe−amorphous Na_2_SiO_3_−Al_2_O_3_ interfaces as a function of the number of atomic layers.

**Figure 6 materials-15-04415-f006:**
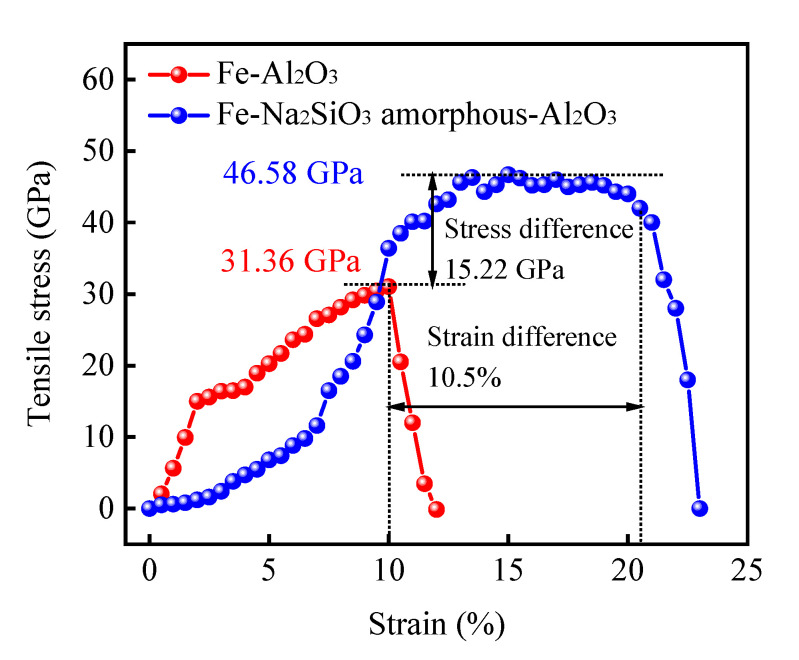
The tensile stress related to strain of the Fe-Al_2_O_3_ and Fe-amorphous Na_2_SiO_3_-Al_2_O_3_ interfaces.

**Figure 7 materials-15-04415-f007:**
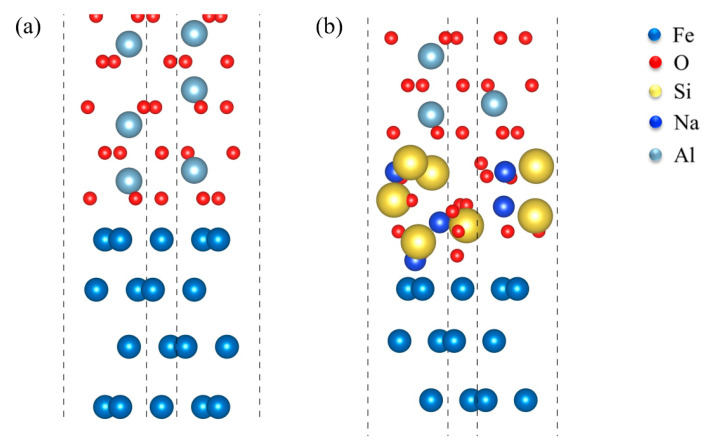
The geometrical structure of the (**a**) Fe-Al_2_O_3_ and the (**b**) Fe-amorphous Na_2_SiO_3_-Al_2_O_3_ interfaces.

**Figure 8 materials-15-04415-f008:**
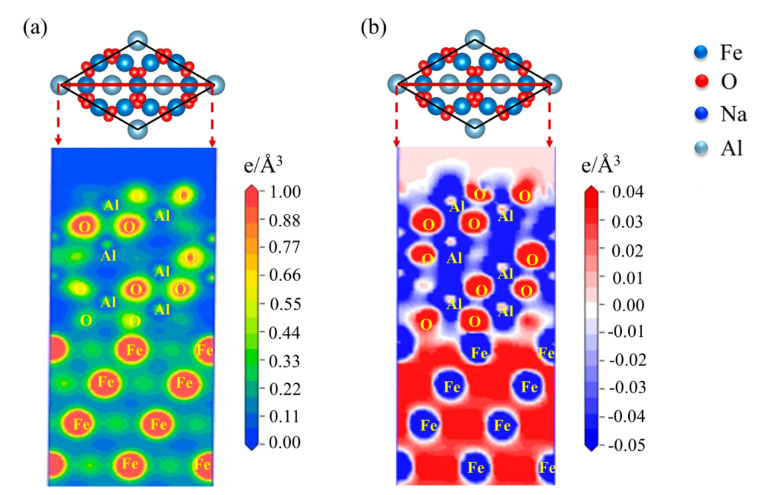
The (**a**) charge density and (**b**) charge density difference of the Fe−Al_2_O_3_ interface at (111) crystal plane.

**Figure 9 materials-15-04415-f009:**
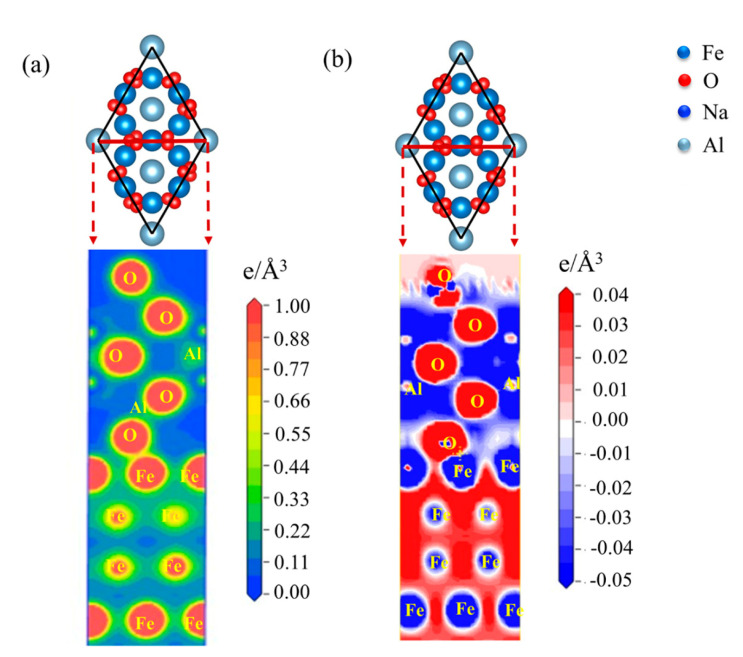
The (**a**) charge density and (**b**) charge density difference of the Fe-Al_2_O_3_ interface at (001) crystal plane.

**Figure 10 materials-15-04415-f010:**
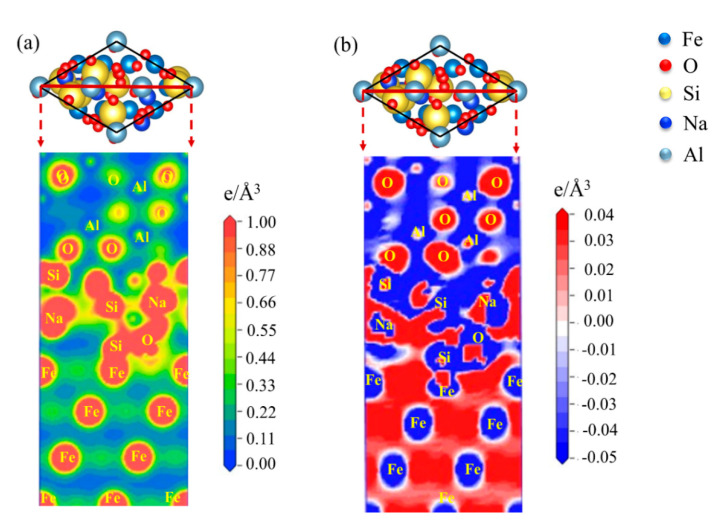
The (**a**) charge density and (**b**) charge density difference of the Fe−amorphous Na_2_SiO_3_−Al_2_O_3_ interface at (111) crystal plane.

**Figure 11 materials-15-04415-f011:**
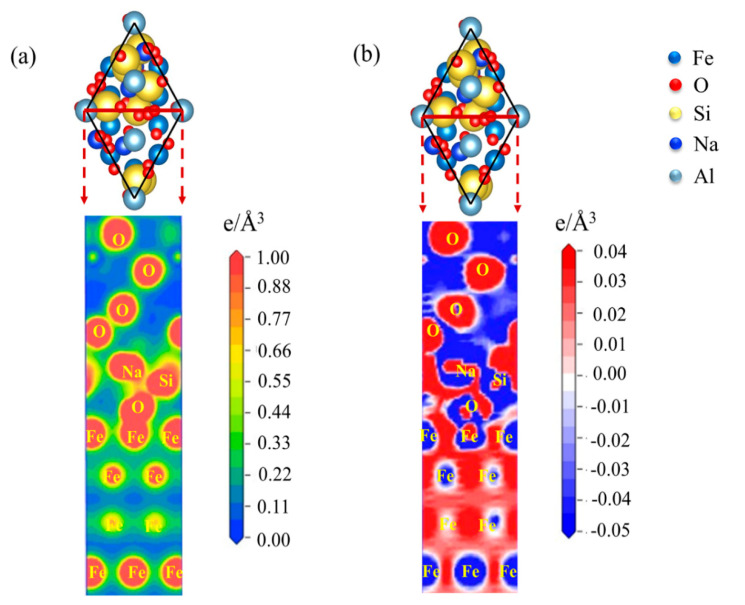
The (**a**) charge density and (**b**) charge density difference of the Fe-amorphous Na_2_SiO_3_-Al_2_O_3_ interface at (001) crystal plane.

**Figure 12 materials-15-04415-f012:**
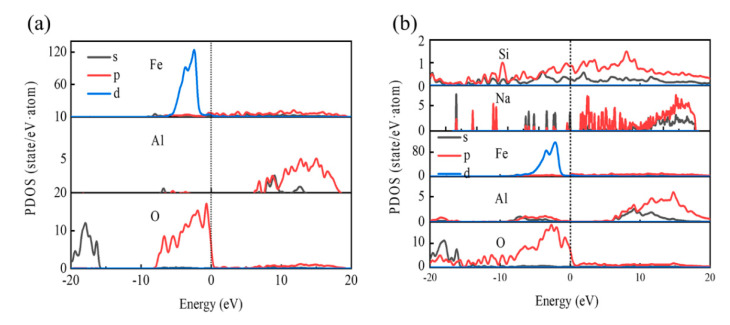
The partial density of states (PDOS) for the (**a**) Fe-Al_2_O_3_ and (**b**) Fe−amorphous Na_2_SiO_3_-Al_2_O_3_ interfaces.

**Table 1 materials-15-04415-t001:** The lattice constants of the *γ*-Fe, Na_2_SiO_3_, and *α*-Al_2_O_3_ bulks.

	*a*	*b*	*c*	*α*	*β*	*γ*
γ-Fe	3.439	3.438	3.438	90°	90°	90°
Na_2_SiO_3_	6.027	10.487	4.784	90°	90°	90°
α-Al_2_O_3_	4.814	4.814	13.156	90°	90°	90°

## Data Availability

The data that support the findings of this study are available upon reasonable request from the authors.

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
