# Peer review of "First-Principles Calculation of the Bonding Strength of the Al2O3-Fe Interface Enhanced by Amorphous Na2SiO3"

_materials, 2022, doi:10.3390/ma15134415_

Round 1

Reviewer 1 Report

This work provided a theoretical explanation for the effect of amorphous sodium metasilicate on the bonding strength of the Fe-Al2O3 interface. The manuscript is fairly well written and presented, with interesting results. I would recommend it for publication after minor revisions. I only have a few questions and comments, as follows:

1.    The authors refer to the interface between Fe and Al2O3, but perhaps should mention why is it interesting to study such interface. Is it related to the use of alumina-reinforced iron-based alloys (i.e. steels)? It would make the Introduction clearer if the target composite systems were specified.

2.    In line 38, the authors mention ‘substrate’ but in a composite system (i.e. alumina dispersed in Fe) it is more correct to refer to it as ‘matrix’.

3.    Why did the authors select γ-Fe structure for the model creation? Wouldn’t α-Fe be more interesting since most Fe-based alloys are ferritic?

4.    In equation 1, it is not clear what E2 and E1/2 are.

5.    In line 123, the authors refer to stress as ‘s’ and strain as ‘e’, but in the formulas 2 and 3 these entities are referred to as sigma (σ) and epsilon (ε), respectively.

6.    In line 185, the authors state that “the charge accumulation between Fe and Al-terminated is weak”, but I could not see in which figure is such model depicted (with Al-terminated interface). Figures 8 and 9 present only O-terminated models, for the Fe-Al2O3 interface.

7.    Have these results been validated by experimental work?

8.    The references list should include article titles.

9.    I recommend proofreading the manuscript.

Author Response

Dear Editor,Please see the attachment.

Author Response

Dear Editor,Please see the attachment.
